# Clinical Characteristics and Outcomes in 314 Japanese Patients with Bacterial Endophthalmitis: A Multicenter Cohort Study from J-CREST

**DOI:** 10.3390/pathogens10040390

**Published:** 2021-03-24

**Authors:** Hiroto Ishikawa, Kazutaka Uchida, Yoshio Takesue, Junya Mori, Takamasa Kinoshita, Shohei Morikawa, Fumiki Okamoto, Tomoko Sawada, Masahito Ohji, Takayuki Kanda, Masaru Takeuchi, Akiko Miki, Sentaro Kusuhara, Tetsuo Ueda, Nahoko Ogata, Masahiko Sugimoto, Mineo Kondo, Shigeo Yoshida, Tadahiko Ogata, Kazuhiro Kimura, Yoshinori Mitamura, Tatsuya Jujo, Hitoshi Takagi, Hiroto Terasaki, Taiji Sakamoto, Takaaki Sugisawa, Yuki Komuku, Fumi Gomi

**Affiliations:** 1Department of Ophthalmology, Hyogo College of Medicine, Nishinomiya 6638501, Japan; sugiwombat@gmail.com (T.S.); yuki.kom0923@gmail.com (Y.K.); gomi.fumi@gmail.com (F.G.); 2Clinical Epidemiology, Hyogo College of Medicine, Nishinomiya 6638501, Japan; kuchida@hyo-med.ac.jp; 3Department of Infection Control and Prevention, Hyogo College of Medicine, Nishinomiya 6638501, Japan; takesuey@hyo-med.ac.jp; 4Department of Ophthalmology, Sapporo City General Hospital, Sapporo 0608604, Japan; junya-mori@doc.city.sapporo.jp (J.M.); knst129@gmail.com (T.K.); 5Department of Ophthalmology, Faculty of Medicine, University of Tsukuba, Tsukuba 3058576, Japan; s0711715@yahoo.co.jp (S.M.); fumiki-o@md.tsukuba.ac.jp (F.O.); 6Department of Ophthalmology, Shiga University of Medical Science, Otsu 5202192, Japan; tsawada@belle.shiga-med.ac.jp (T.S.); ohji@belle.shiga-med.ac.jp (M.O.); 7Department of Ophthalmology, National Defense Medical College, Tokorozawa 3598513, Japan; kankan@ndmc.ac.jp (T.K.); masatake@ndmc.ac.jp (M.T.); 8Department of Surgery, Division of Ophthalmology, Kobe University Graduate School of Medicine, Kobe 6500017, Japan; acacyey@med.kobe-u.ac.jp (A.M.); kusu@med.kobe-u.ac.jp (S.K.); 9Department of Ophthalmology, Nara Medical University School of Medicine, Kashihara 6348522, Japan; uedatetuo@hotmail.com (T.U.); ogata@naramed-u.ac.jp (N.O.); 10Department of Ophthalmology, Mie University Graduate School of Medicine, Tsu 5148507, Japan; sugmochi92@gmail.com (M.S.); mineo@clin.medic.mie-u.ac.jp (M.K.); 11Department of Ophthalmology, Kurume University School of Medicine, Kurume 8300011, Japan; yoshi@med.kurume-u.ac.jp; 12Department of Ophthalmology, Graduate School of Medicine, Yamaguchi University, Ube 7558505, Japan; ogt017@yamaguchi-u.ac.jp (T.O.); k.kimura@yamaguchi-u.ac.jp (K.K.); 13Department of Ophthalmology, Tokushima University Graduate School, Tokushima 7708503, Japan; ymitaymitaymita@yahoo.co.jp; 14Department of Ophthalmology, St. Marianna University School of Medicine, Kawasaki 2168511, Japan; t2jujo@marianna-u.ac.jp (T.J.); htakagi@marianna-u.ac.jp (H.T.); 15Department of Ophthalmology, Kagoshima University Graduate School of Medical and Dental Sciences, Kagoshima 8908520, Japan; hirototerasaki112@gmail.com (H.T.); tsakamot@m3.kufm.kagoshima-u.ac.jp (T.S.)

**Keywords:** endophthalmitis, exogenous endophthalmitis, endogenous endophthalmitis, retrospective study, vitrectomy, visual acuity

## Abstract

Bacterial endophthalmitis is an intraocular infection that causes rapid vison loss. Pathogens can infect the intraocular space directly (exogenous endophthalmitis (ExE)) or indirectly (endogenous endophthalmitis (EnE)). To identify predictive factors for the visual prognosis of Japanese patients with bacterial endophthalmitis, we retrospectively examined the bacterial endophthalmitis characteristics of 314 Japanese patients and performed statistics using these clinical data. Older patients, with significantly more severe clinical symptoms, were prevalent in the ExE group compared with the EnE group. However, the final best-corrected visual acuity (BCVA) was not significantly different between the ExE and EnE groups. Bacteria isolated from patients were not associated with age, sex, or presence of eye symptoms. *Genus Streptococcus*, *Streptococcus pneumoniae,* and *Enterococcus* were more prevalent in ExE patients than EnE patients and contributed to poor final BCVA. The presence of eye pain, bacterial identification, and poor BCVA at baseline were risk factors for final visual impairment.

## 1. Introduction

Bacterial endophthalmitis is a purulent inflammation of the intraocular fluids, i.e., the vitreous and the aqueous humor that leads to significant visual loss with progression in hours to days. The foundation of the treatment for bacterial endophthalmitis is focal and systemic administration of antibiotics; vitrectomy is performed if the endophthalmitis is severe. Unless the patient receives appropriate treatment early in the disease, it causes irreversible damage to the retina, which results in a high probability of vision loss [1,2,3,4,5,6,7,8].

Bacterial endophthalmitis is classified into two groups on the basis of the infection route (exogenous or endogenous). Exogenous endophthalmitis (ExE) is mainly caused by eye surgery [9,10,11,12,13,14,15,16,17,18,19,20,21,22,23] and trauma [24,25,26] due to the direct infection of pathogens into the eye from outside. Acute endophthalmitis appears as a postoperative complication within 1–2 weeks after the surgical intervention, most commonly on the third to fifth postoperative day [1,2]. In endogenous endophthalmitis (EnE), there is an infection focus (or foci) somewhere else in the body, and then the pathogens are transferred to the eye [27,28,29,30,31,32,33,34].

Several types of causative pathogens have been reported, with a trend to different types of causative pathogens for ExE and EnE. In eyes with ExE, the normal conjunctival florae are the typical causative pathogens [35,36,37,38]. Conversely, bacteremia can cause EnE.

Despite recent advances in the science and technology for the identification of pathogens using polymerase chain reaction (PCR) [39,40] and matrix-assisted laser desorption/ionization time-of-flight mass spectrometry [41,42], new antibiotics, and minimally invasive vitrectomy, bacterial endophthalmitis remains a severe intraocular infection that causes a high risk of vision loss. Recently, a large cohort study in Japanese patients with postoperative endophthalmitis after cataract surgery showed that the incidence of endophthalmitis was 0.025% [12]. In the present study, we retrospectively examined the characteristics of 314 bacterial endophthalmitis cases seen in multiple institutions of the Japan Clinical REtina STudy group (J-CREST) in a 10-year span and performed statistics to identify predictive factors for visual prognosis using these clinical data.

## 2. Results

### 2.1. Patients’ Characteristics

The patients’ characteristics are shown in Table 1. Briefly, 278 patients (88.5%) developed infection unilaterally, and 242 eyes (69.1%) developed ExE. Additionally, in 84.2% and 15.8% of unilateral cases, the patients developed ExE and EnE, respectively. Bacteria were identified in 117 eyes (37.7%). Vitrectomy was performed in 276 eyes (79.1%) as a primary treatment. Overall, BCVA had significantly improved 3 months after treatment (*p* < 0.0001).

### 2.2. Better Vision Group versus Legal Blindness Group

We performed univariate and multivariate analyses for contributions of nine factors: age, sex, right or left eye, ExE or EnE, duration from the onset (days), presence of eye symptoms (eye pain and ciliary injection), bacterial identification, and initial BCVA (Table 2).

Univariate analysis showed that four factors, presence of eye pain (*p* < 0.0001), presence of ciliary injection (*p* = 0.01), bacterial identification (*p* < 0.0001), and poor initial BCVA (*p* < 0.0001), significantly contributed to legal blindness. Multivariate analysis of these four factors showed that three factors (presence of eye pain, bacterial identification, and poor BCVA at the baseline) significantly contributed to the legal blindness group (Table 2).

### 2.3. Primary Endpoint: Contribution of Initial BCVA to Visual Prognosis

The receiver operating characteristic (ROC) curve of the better vison group, as shown in Figure 1, suggests that the area under the curve (AUC) for the ROC curve was 0.83. The cut-off value was 1.7 logMAR (Snellen ≥ 6/300).

### 2.4. Secondary Endpoints: Exogenous versus Endogenous Endophthalmitis

To investigate the difference between ExE and EnE, we compared the characteristics in the two groups (Table 3). Briefly, the ExE patients were older than the EnE patients (*p* = 0.0004). Significantly more of the ExE patients presented with severe clinical symptoms, including eye symptoms (eye pain and ciliary injection), and the ExE patients had worse initial BCVA than did the EnE patients (*p* < 0.0001, *p* = 0.0001, respectively). However, the final BCVAs between the two groups did not differ significantly.

### 2.5. Secondary Endpoints: Analyses of pathogens

We classified the pathogens into eight types: *coagulase-negative staphylococci* (CoNS), *Staphylococcus aureus*, *Genus Streptococcus*, *Streptococcus pneumoniae*, *Enterobacteriaceae*, *Enterococcus*, other bacteria, and fungus. Characteristic associated with the eight types are shown in Table 4. Briefly, age, sex and presence of eye symptoms (eye pain and ciliary injection) did not show any significant differences between the eight types. CoNS, *Streptococcus pneumoniae*, *Enterococcus,* and other bacteria were detected at significantly higher rates in patients with ExE than in those with EnE. The best initial BCVAs were seen in patients with fungus, and the worst final BCVAs were seen in patients with *Genus Streptococcus*, *Streptococcus pneumoniae,* or *Enterococcus*.

### 2.6. Secondary Endpoints: Analyses of Selected Treatments

To investigate the differences between the therapy types, we determined the characteristics in three groups, as shown in Table 5. Briefly, vitrectomy was performed in older patients, and in eyes with ExE, shorter duration from the onset, eye pain, ciliary injection, bacterial identification, and poor initial BCVA, which is consistent with the consensuses of retinal surgeons in J-CREST. Eyes that needed only antibiotic eye drops and/or intravitreal antibiotics and/or systemic antibiotics had a very good chance of achieving good final vision.

## 3. Discussion

We showed here the characteristics of 314 Japanese patients (350 eyes) with bacterial endophthalmitis. Ocular pain, bacterial identification, and poor BCVA at baseline contributed to the final visual impairment. In particular, if the initial BCVA was under 1.7 logMAR (Snellen ≥ 6/300), the final BCVA was likely to stay below 1.0 logMAR (Snellen 6/60 or better).

Our results were similar to several reports in the literature from other countries regarding the rate of bacterial identification (our results: 37.7% by culturing and PCR, 39.7% by culturing [7], 43.4% by culturing [23], 23% by culturing [31], 28.6% by culturing [33], 38.6% by PCR [39], 30% by culturing [40], and 45.5% by culturing [42]). PCR has become a commonly used tool for the identification of ocular pathogens [39,40], but the identification of bacteria is still difficult. However, we showed that the presence of identifiable bacteria is one of the risk factors for final visual impairment, which suggests that substantial amounts of intraocular pathogens are needed for bacterial detection. As PCR technology progresses, the rate of bacterial identification becomes much higher; with further progress, bacterial identification, which the present study showed was a risk factor for final visual impairment, will no longer be a risk factor.

BCVA in patients with endophthalmitis improved significantly after treatment, and the percentage of patients with final BCVA ≤ 1.0 logMAR was 76.6%, which suggests that our treatments for endophthalmitis in J-CREST were not inferior to those in other reports (33.3% (Snellen 20/31 or better) [5], 28% (Snellen 20/120 or better) [6], 44% (Snellen 20/40 or better) [18], 32.7% (Snellen 20/40 or better) [22], 71% (Snellen 5/200 or better) [43], 34.5% (Snellen 20/400 or better) [44], and 34.5% (Snellen 20/60 or better) [45]). Additionally, the ophthalmologists in J-CREST (all are retina specialists) tended to perform a vitrectomy as a primary treatment for endophthalmitis, which is likely a worldwide trend [4,24,46,47].

In the present analyses, three factors (presence of eye pain, bacterial identification, and poor initial BCVA) significantly contributed to legal blindness, which suggests that a severe condition at baseline is a risk factor for later visual impairment. This is not very surprising, as the same findings appear in the literature: eye pain [11], bacterial identification [29,48,49], and poor initial BCVA [11,25,26,29,30,33,50]. Other factors that have been reported as risk factors include: younger age (<85 years) [21], female sex [30], presence of an intraocular foreign body [25], a higher number of intravitreal injections [25,30], the type of injury (rupture) [26], retinal detachment [26,48], and proliferative vitreoretinopathy [26]. The most important risk factor for visual impairment at 12 weeks was poor initial BCVA. The present study showed that the final BCVA could remain under 1.0 logMAR when the baseline BCVA was under 1.7 logMAR (Snellen 6/300 or better). This cut-off value is suggested to be a good parameter for informed consent and for prognosis of final BCVA in patients with endophthalmitis.

In this study, there were 242 eyes (69.1%) with ExE and 108 eyes (30.9%) with EnE. From the literature, the rates of ExE are 85.0% [5], 92.2% [7], 82.8% (*Enterococcus* endophthalmitis) [44], and 78.4% [46]. Thus, the present study showed an EnE rate that was higher than the results in the literature. ExE was observed in older patients and was significantly associated with presentation of severe clinical symptoms. However, the final BCVAs between the two groups were not significantly different. In developing countries, ocular trauma is the main cause of ExE [7]. Conversely, in the developed countries, ExE is mainly caused by ocular surgeries, such as cataract surgery, so there are much older patients with ExE than with EnE.

Patients with ExE after ocular surgeries or trauma showed rapid worsening of eye conditions, due to the direct invasion of pathogens. Therefore, the infection speed in the eyeball can be faster than it is with EnE. The bacterial growth corresponds with the severe clinical symptoms. Although the final BCVA was not significantly different between the ExE and EnE groups, patients with EnE often have poor general health condition, e.g., immunosuppressive state, a terminal stage of cancer, and cannot communicate well with medical staff, so the clinical symptoms may simply appear milder than in ExE. Additionally, because of the rapid worsening of eye conditions in eyes with ExE, patients with ExE may consult a doctor sooner than do patients with EnE, which suggests one possible reason for the lack of difference in final BCVA between ExE and EnE patients. Moreover, we sometimes had cases in which it was difficult to distinguish between noninfectious uveitis and EnE. We, as ophthalmologists, should keep the possibility of EnE in mind.

In eyes with EnE, the pathogens are carried by the blood stream transarterially. Therefore, we think that infection is more common in the left eye than in the right eye, but the present study did not prove that.

Regarding the types of pathogens observed, CoNS, *Enterococcus*, *Streptococcus pneumoniae,* and other bacteria were detected significantly more commonly in patients with ExE than EnE. CoNS, *Enterococcus*, and *Streptococcus pneumoniae* are normal conjunctival florae, and their detection rate increases with patient age [35,36,37,38]. The best baseline BCVA was seen in patients with fungal infection, and poorest final BCVA was seen in patients with *Genus Streptococcus*, *Streptococcus pneumoniae,* and *Enterococcus*. Fungal endophthalmitis is usually detectable at an early stage in hospitals, so the clinical symptoms at baseline may be mild. Conversely, *Enterococcus*, *Streptococcus pneumoniae,* and *Genus Streptococcus* are highly virulent, which suggests that the ocular tissue is easily harmed. Therefore, we must rapidly make these diagnoses.

Vitrectomy was performed in eyes with severe infection. It is reasonable for any ophthalmologist to treat endophthalmitis. Studies have concluded that vitrectomy is effective for the initial treatment of endophthalmitis; early vitrectomy (i.e., within 24 h) might be helpful in achieving a better visual outcome [2,4,46,51], and a primary vitrectomy demonstrated greater visual improvement [2,18,47]. However, the European Vitreo-Retinal Society Endophthalmitis Study showed similar outcomes after treatment with intravitreal antibiotics alone compared with intravitreal antibiotics and vitrectomy within 1 week of presentation [45]. Interestingly, the present study showed that 94% of eyes without intravitreal antibiotics or vitrectomy had final BCVA under 1.0 logMAR, which suggests that J-CREST retina specialists were able to decide the appropriate treatments, depending on the disease severity.

We acknowledge several limitations to the present study. First, the bacterial identification rates may have differed between centers because the companies that performed the tests were not standardized, i.e., different laboratories were used by the different hospitals in this multicenter retrospective study. Second, the follow-up duration was only 3 months, so the results may change when we examine data for longer follow-up periods.

In summary, we retrospectively examined the characteristics of Japanese patients with endophthalmitis and showed that eye pain, bacterial identification, and poor BCVA at baseline were the three risk factors for visual impairment at 12 weeks. Notably, the final BCVA could remain under 1.0 logMAR when the baseline BCVA was under 1.7 logMAR (Snellen ≥ 6/300). Endophthalmitis remains a severe ocular infection, but we can challenge it with rapid and bold treatments, with advances in medical knowledge and technology.

## 4. Materials and Methods

### 4.1. Study Design and Eligibility

This was a multicenter, retrospective study of consecutive patients with bacterial endophthalmitis in the following 13 J-CREST institutions: Hyogo College of Medicine, Sapporo City General Hospital, the University of Tsukuba, Shiga University of Medical Science, the National Defense Medical College, Kobe University Graduate School of Medicine, Nara Medical University School of Medicine, Mie University, Kurume University School of Medicine, Yamaguchi University, Tokushima University, St. Marianna University School of Medicine, and Kagoshima University Graduate School of Medical and Dental Sciences, between January 2010 and November 2019, inclusive. The data, analytic methods, and study materials will be made available to other researchers for purposes of reproducing the results.

### 4.2. Patients

At each hospital in the J-CREST group, data from patients with bacterial endophthalmitis were analyzed using medical records. The bacterial endophthalmitis was judged by a retinal specialist in each hospital to have a “definite” diagnosis (could prove pathogens) or “a strong suspicion” (could not prove pathogens). The observation period was >12 weeks for all analyzed subjects. Among the 325 Japanese patients (364 eyes) with bacterial endophthalmitis initially enrolled in the present study, data from 11 patients were excluded from the analysis because of a lack of follow-up at 12 weeks. Therefore, a total of 350 eyes in 314 patients were analyzed.

### 4.3. Protocol

Data were extracted from medical records in the various hospitals and sent to the data center in the Department of Ophthalmology, Hyogo College of Medicine. We analyzed patient age, sex, type of bacterial endophthalmitis, duration from the onset of symptoms to the initial treatment, main complaint (presence of ocular pain and ciliary injection), whether bacteria were identified, details of identified bacteria, and best-corrected visual acuity (BCVA) at baseline and 4 and 12 weeks after treatment started. BCVA was measured on a Japanese Snellen chart, with values converted to log minimum angle of resolution (logMAR). For analysis, patients were grouped on the basis of BCVA at 12 weeks. Those with BCVA ≤ 1.0 logMAR (Snellen 6/60 or better) are referred to as the “better vision group” versus those with BCVA > 1.0 logMAR (Snellen < 6/60) are the “legal blindness” group [52].

### 4.4. Endpoints

The primary endpoint was to assess the contribution of initial BCVA to visual prognosis. Therefore, we examined the difference between the patients in the better vision and legal blindness group at 12 weeks. We also examined the difference between ExE and EnE and determined pathogens that contributed to poor BCVA. Additionally, this study compared between patients in the better vision and legal blindness groups. Risk factors for the legal blindness group were investigated based on the patients’ characteristics, eye symptoms, and initial BCVA.4.5. Statistical Analyses

Categorical variables are presented as numbers and percentages and compared using the Chi-square test. Continuous variables are expressed as the mean ± standard deviation (SD), or the median and interquartile range, and compared using a *t*-test or the Wilcoxon rank-sum test. We constructed a multivariable logistic model to explore the variables associated with BCVA at 12 weeks. Logistic regression analysis was performed to identify potential risk factors underlying BCVA at 12 weeks, incorporating all variables with a *p*-value < 0.01 in the univariate analysis. Uncertain or undetectable data were considered null, and missing values for laboratory data were eliminated from the analyses. All statistical analyses were conducted using JMP 14.2.0 (SAS Institute Inc., Cary, NC). All reported *p*-values were two-tailed, and the statistical significance level was set at *p* < 0.05.

## Figures and Tables

**Figure 1 pathogens-10-00390-f001:**
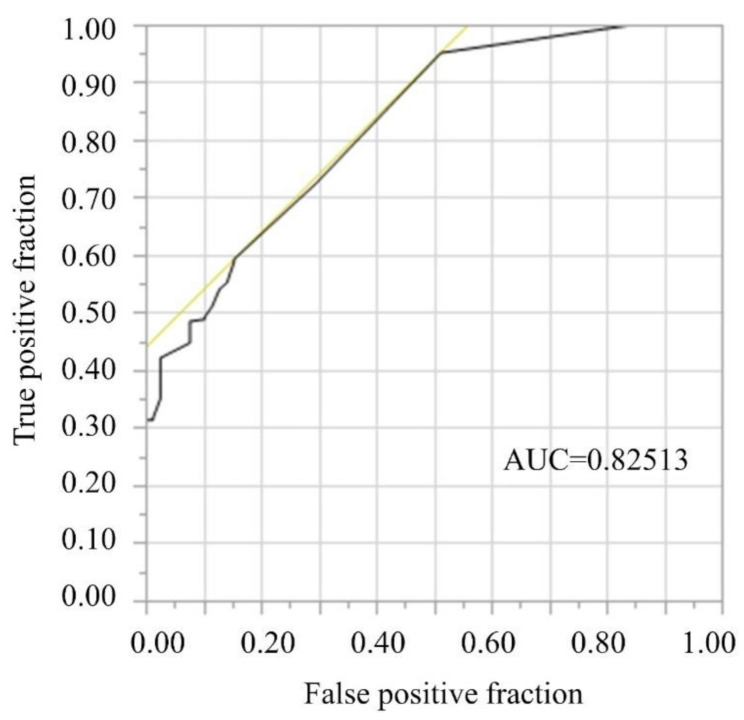
The receiver operating characteristic (ROC) curve of the better vison group. The area under the curve (AUC) for the ROC curve was 0.83. The cut-off value was 1.7 logMAR (Snellen ≥ 6/300).

**Table 1 pathogens-10-00390-t001:** Patients’ characteristics.

Variables	Mean ± Standard Deviation or Number/Total (%)
Age, years	68.3 ± 15.3
Men	154/314 (49.0%)
Unilateral cases	278/314 (88.5%)
Left eye	200/350 (57.1%)
Exogenous endophthalmitis	242/350 (69.1%)
Duration from the onset to the initial treatment, days	5.4 ± 18.2 (0–298)
Eye pain	148/347 (42.7%)
Ciliary injection	182/349 (52.1%)
Bacteria identified	117/310 (37.7%)
Vitrectomy	276/349 (79.1%)
Initial BCVA ^a^ (logMAR ^b^)	1.48 ± 0.96
Final BCVA (logMAR)	0.72 ± 0.93
Final BCVA ≤ 1.0 logMAR (Snellen 6/60 or better)	268/350 (76.6%)
Enucleation	21/350 (6.0%)

^a^ BCVA, best-corrected visual acuity. ^b^ logMAR, log (minimum angle of resolution).

**Table 2 pathogens-10-00390-t002:** Characteristics of patients in the “better vision” and “legal blindness” groups.

Variables	Final BCVA ^a^ ≤ 1.0 logMAR ^b^ (Snellen 6/60 or Better) (Better Vison Group)	Final BCVA > logMAR (Snellen < 6/60) (Social Blindness Group)	*p*-Values; Univariate	*p*-Values; Multivariate
Age, years	67.7 ± 15.2	69.7 ± 15.1	0.24	
Men	131/271 (48.3%)	38/79 (48.1%)	1.00	
Left eye	149/271 (55.0%)	51/79 (64.1%)	0.16	
Exogenous endophthalmitis	186/271 (68.6%)	56/79 (70.9%)	0.78	
Duration from the onset to the initial treatment, days	4.6 ± 8.6	8.1 ± 34.5	0.23	
Eye pain	94/268 (35.1%)	54/79 (68.4%)	***<0.0001***	***0.002***
Ciliary injection	131/270 (48.5%)	51/79 (64.6%)	***0.01***	0.25
Bacterial identification	73/233 (31.3%)	44/77 (57.1%)	***<0.0001***	***0.04***
Initial BCVA (logMAR)	1.25 ± 0.93	2.27 ± 0.58	***<0.0001***	***<0.0001***

^a^ BCVA, best-corrected visual acuity. ^b^ logMAR, log (minimum angle of resolution). Bold italics indicate statistical significance (*p* < 0.05).

**Table 3 pathogens-10-00390-t003:** Characteristics of patients in the exogenous endophthalmitis (ExE) and endogenous endophthalmitis (EnE) groups.

Variables	ExE	EnE	*p*-Values
Age, Years	69.7 ± 15.2	64.8 ± 14.5	***0.0004***
Men	116/242 (47.9%)	53/108 (49.1%)	0.91
Left Eye	145/242 (59.9%)	55/108 (50.9%)	0.13
Duration from the Onset to the Initial Treatment, Days	4.6 ± 20.2	7.8 ± 11.3	***<0.0001***
Eye Pain	123/240 (51.3%)	25/107 (23.4%)	***<0.0001***
Ciliary Injection	146/241 (60.6%)	36/108 (33.3%)	***<0.0001***
Bacterial Identification	82/236 (34.8%)	35/74 (47.3%)	0.056
Initial BCVA ^a^ (logMAR ^b^)	1.61 ± 0.90	1.18 ± 1.04	***0.0001***
Final BCVA ≤ 1.0 logMAR (Snellen 6/60 or Better)	183/238 (76.9%)	85/108 (78.7%)	0.79

^a^ BCVA, best-corrected visual acuity. ^b^ logMAR, log (minimum angle of resolution). Bold italics indicate statistical significance (*p* < 0.05).

**Table 4 pathogens-10-00390-t004:** Characteristics of patients grouped by bacterial types.

Variables	CoNS ^a^	*S. aureus* ^b^	Genus *Streptococcus*	*S. pneumoniae* ^c^	*Enterobacteriaceae*	*Enterococcus*	Other Bacteria	Fungus	*p*-Values
Age, years	67.5 ± 2.8	67.5 ± 4.2	70.4 ± 4.8	63.3 ± 7.1	68.9 ± 6.5	75.8 ± 8.6	60.5 ± 5.0	70.0 ± 3.9	0.97
Men	21/39 (53.9%)	8/17 (47.1%)	5/13 (38.5%)	4/6 (66.7%)	3/7 (42.9%)	1/4 (25.0%)	7/12 (58.3%)	11/20 (55.0%)	0.88
Exogenousendophthalmitis	39/39 (100%)	10/17 (58.8%)	6/13 (46.2%)	5/6 (83.3%)	1/7 (14.3%)	4/4 (100%)	11/12 (91.7%)	7/20 (35.0%)	***<0.0001***
Eye pain	21/38 (55.3%)	9/17 (52.9%)	9/13 (69.2%)	5/6 (83.3%)	4/7 (57.1%)	2/4 (50.0%)	7/12 (58.3%)	5/20 (25.0%)	0.17
Ciliary injection	20/38 (52.6%)	11/17 (64.7%)	10/13 (76.9%)	4/6 (66.7%)	4/7 (57.1%)	2/4 (50.0%)	10/12 (83.3%)	10/20 (50.0%)	0.48
Initial BCVA ^d^ (logMAR ^e^)	1.84	2.04	2.47	2.45	2.51	2.60	1.86	1.11	***<0.0001***
Final BCVA ≤ 1.0 logMAR (Snellen 6/60 or better)	34/39 (86.5%)	12/17 (70.6%)	4/13 (30.8%)	1/6 (16.7%)	3/7 (42.9%)	0/4 (0%)	7/12 (58.3%)	13/20 (65.0%)	***<0.0001***

^a^ CoNS, *Coagulase-negative staphylococci*. ^b^
*S. aureus*, *Staphylococcus aureus*. ^c^
*S. pneumoniae*, *Streptococcus pneumoniae*. ^d^ BCVA, best-corrected visual acuity. ^e^ logMAR, log (minimum angle of resolution). Bold italic indicates statistical significance (*p* < 0.05).

**Table 5 pathogens-10-00390-t005:** Patients’ characteristics according to therapy types.

Variables	(1) Antibiotics Administration via Eye Drops and/or Systemic (*n* = 52)	(2) Intravitreal Antibiotics and/or (1) (*n* = 21)	(3) Vitrectomy and/or (2) (*n* = 276)	*p*-Values
Age, Years	62.5 ± 16.3	70.6 ± 12.7	69.0 ± 15.0	***0.010***
Men	25/52 (48.1%)	12/21 (57.1%)	132/276 (47.8%)	0.74
Left Eye	27/52 (51.9%)	14/21 (66.7%)	159/276 (57.6%)	0.51
ExE ^a^	17/52 (32.7%)	15/21 (71.4%)	209/276 (75.7%)	***<0.0001***
Duration from the Onset to the Initial Treatment, Days	6.2 ± 9.1	7.8 ± 9.9	5.1 ± 19.6	***0.019***
Eye Pain	5/51 (9.8%)	10/21 (47.6%)	132/274 (48.2%)	***<0.0001***
Ciliary Injection	11/52 (21.2%)	5/21 (23.8%)	165/275 (60.0%)	***<0.0001***
Bacterial Identification	1/21 (4.8%)	3/17 (17.7%)	113/271 (41.7%)	***0.0002***
Initial BCVA ^b^ (logMAR ^c^)	0.39 ± 0.71	1.10 ± 1.10	1.71 ± 0.84	***<0.0001***
Final BCVA ≤ 1.0 logMAR (Snellen 6/60 or better)	47/50 (94.0%)	14/20 (70.0%)	206/275 (77.4%)	***0.003***

^a^ ExE, exogenous endophthalmitis. ^b^ BCVA, best-corrected visual acuity. ^c^ logMAR, log (minimum angle of resolution). Bold italics indicate statistical significance (*p* < 0.05).

## Data Availability

The data presented in this study are available in Appendix A.

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
