# Peer review of "Clinical Characteristics and Outcomes in 314 Japanese Patients with Bacterial Endophthalmitis: A Multicenter Cohort Study from J-CREST"

_pathogens, 2021, doi:10.3390/pathogens10040390_

Round 1
Reviewer 1 Report
Although similar studies have been done before by many group but in this comprehensive well-designed statistical study from JCREST, Ishikawa et al have shown patients with initial poor BCVA above 1.7 logMAR had severe ExE, they also indicated eye pain and bacterial identification as an additional risk factor for ExE progression prediction. The manuscript is written well with enough background for the reader, statistical and method is well explained, and the data supports the conclusion drawn. Authors have also discussed the main weakness of the study i.e, varied bacterial identification rate & methods which is understandable for a large multi-cohort study and the patients were followed for short duration (12 weeks). Although longer follow-up study would have been more valuable as most of the patient age 68.3±15.3 years.
Few minor concerns:
- Authors can replace the “real-world” with clinical data which seems to be more appropriate (line 47, 82) also might replace the word “developing/developed countries” with “other countries” (line164,203 etc).
- Might change the science & technology instead (line 73)
- Need to italicized all full bacterial scientific names (line130, etc)
- In the result section 2.4 instead of “Other end-points:” authors can mention as secondary end-points (line128,145 also).
Author Response
Reviewer 1:
Although similar studies have been done before by many group but in this comprehensive well-designed statistical study from JCREST, Ishikawa et al have shown patients with initial poor BCVA above 1.7 logMAR had severe ExE, they also indicated eye pain and bacterial identification as an additional risk factor for ExE progression prediction. The manuscript is written well with enough background for the reader, statistical and method is well explained, and the data supports the conclusion drawn. Authors have also discussed the main weakness of the study i.e, varied bacterial identification rate & methods which is understandable for a large multi-cohort study and the patients were followed for short duration (12 weeks). Although longer follow-up study would have been more valuable as most of the patient age 68.3±15.3 years.
Response
Thank you for your affirmative opinions.
Few minor concerns:
Authors can replace the “real-world” with clinical data which seems to be more appropriate (line 47, 82) also might replace the word “developing/developed countries” with “other countries” (line164,203 etc).
Response
As requested, the phrases of “real-world data” were changed to “clinical data”.
Also, “developed countries” in line 164 was changed to “other countries”.
However, “developed” in line 203, and “developing” in line 204 were not changed because we contrasted characteristics of endophthalmitis between developed and developing countries.
Might change the science & technology instead (line 73)
Response
As requested, we replaced the description.
Need to italicized all full bacterial scientific names (line130, etc)
Response
We italicized all full bacterial scientific name throughout the main text.
In the result section 2.4 instead of “Other end-points:” authors can mention as secondary end-points (line128,145 also).
Response
As requested, we replaced the description.
Reviewer 2 Report
Dear Authors,
I find your paper very interesting. However, some issue need to be adressed.
I suggest to improve the Introduction.
First, it should provide a wider overview regarding the percentage of incidence and the different types of onset. Acute endophthalmitis appears as a postoperative complication within 1–2 weeks after the surgical intervention, most commonly on the third to fifth postoperative day. Please add the results related to the onset of endophthalmitis in your cohort if you have. Please cite the following articles*:
-*Lalwani, G.A. et al. Acute-onset endophthalmitis after clear corneal cataract surgery (1996–2005). Clinical features, causative organisms, and visual acuity outcomes. Ophthalmology 2008.
-*Pietras-Baczewska, A. et al. Urgent Vitrectomy with Vancomycin Infusion, Silicone Oil Endotamponade, and General Antibiotic Treatment in Multiple Cases of Endophthalmitis from a Single Day of Intravitreal Injections—Case Series. J. Clin. Med. 2021, 10, 1059.
Endophthalmitis is defined as a critical intraocular infection. Please include a more detailed definition of the disease. It is better defined as is a purulent inflammation of the intraocular fluids, i.e.,the vitreous and the aqueous humor**.
Reference:
-**Pietras-Baczewska, A. et al. Urgent Vitrectomy with Vancomycin Infusion, Silicone Oil Endotamponade, and General Antibiotic Treatment in Multiple Cases of Endophthalmitis from a Single Day of Intravitreal Injections—Case Series. J. Clin. Med. 2021, 10, 1059.
Author Response
Reviewer 2: Dear Authors,
I find your paper very interesting. However, some issue need to be addressed.
I suggest to improve the Introduction.
First, it should provide a wider overview regarding the percentage of incidence and the different types of onset. Acute endophthalmitis appears as a postoperative complication within 1–2 weeks after the surgical intervention, most commonly on the third to fifth postoperative day. Please add the results related to the onset of endophthalmitis in your cohort if you have. Please cite the following articles*:
Response
Thank you for your valuable suggestion. We agreed to describe the different types of onset. We added the description and references as the reviewer requested in the introduction section.
About the data of the onset of endophthalmitis, we already showed it in the table1. Also, we prepared 2nd manuscript about Exogenous endophthalmitis and 3rd manuscript about Endogenous endophthalmitis, so we did not appeal more details of each types in the present manuscript.
Endophthalmitis is defined as a critical intraocular infection. Please include a more detailed definition of the disease. It is better defined as is a purulent inflammation of the intraocular fluids, i.e.,the vitreous and the aqueous humor**.
Response
As requested, we changed the description and added the reference.
Round 2
Reviewer 2 Report
thanks for the answers. I have no addtional comments.